# DIST-CLIP: Arbitrary Metadata and Image Guided MRI Harmonization via Disentangled Anatomy-Contrast Representations

**Mehmet Yigit Avci**[1]                                           YIGIT.AVCI@KCL.AC.UK
**Pedro Borges**[1]                                               PEDRO.BORGES@KCL.AC.UK
**Virginia Fernandez**[1]                                   VIRGINIA.FERNANDEZ@KCL.AC.UK
**Paul Wright**[1]                                                 P.WRIGHT@KCL.AC.UK
**Mehmet Yigitsoy**[2]                                     MEHMET.YIGITSOY@GMAIL.COM
**Sebastien Ourselin**[1]                                   SEBASTIEN.OURSELIN@KCL.AC.UK
**Jorge Cardoso**[1]                                         M.JORGE.CARDOSO@KCL.AC.UK

[1]*School of Biomedical Engineering & Imaging Sciences, King's College London, London, UK*
[2]*deepc GMBH, Munich, Germany*

**Editors:** Accepted for publication at MIDL 2026

## Abstract

Deep learning holds immense promise for transforming medical image analysis, yet its clinical generalization remains profoundly limited. A major barrier is data heterogeneity. This is particularly true in Magnetic Resonance Imaging, where scanner hardware differences, diverse acquisition protocols, and varying sequence parameters introduce substantial domain shifts that obscure underlying biological signals. Data harmonization methods aim to reduce these instrumental and acquisition variability, but existing approaches remain insufficient. When applied to imaging data, image-based harmonization approaches are often restricted by the need for target images (i.e., mapping source to target modality given a reference image), while existing text-guided methods rely on simplistic labels that fail to capture complex acquisition details or are typically restricted to datasets with limited variability (i.e., mapping source to target modality given some conditioning text), failing to capture the heterogeneity of real-world clinical environments. To address these limitations, we propose DIST-CLIP (Disentangled Style Transfer with CLIP Guidance), a unified framework for MRI harmonization that flexibly uses either target images or DICOM metadata for guidance. Our framework explicitly disentangles anatomical content from image contrast, with the contrast representations being extracted using pre-trained CLIP encoders. These contrast embeddings are then integrated into the anatomical content via a novel Adaptive Style Transfer module. We trained and evaluated DIST-CLIP on diverse real-world clinical datasets, and showed significant improvements in performance when compared against state-of-the-art methods in both style translation fidelity and anatomical preservation, offering a flexible solution for style transfer and standardizing MRI data. Our code and weights are publicly available at https://github.com/myigitavci/MaRaI.

**Keywords:** MRI harmonization, Disentanglement, Style Transfer, CLIP, Contrastive Learning

## 1. Introduction

Deep learning models have become central to medical image analysis, demonstrating state-of-the-art performance in tasks such as segmentation, classification, and disease prognosis

(Litjens et al., 2017). However, their clinical translation remain severely limited by their poor generalization (Kelly et al., 2022). Models trained on data from one scanner or institution often experience a significant performance drop when applied to data acquired with different settings (Sendra-Balcells et al., 2022). This domain shift is particularly pronounced in Magnetic Resonance Imaging (MRI), where differences in acquisition settings stored in DICOM metadata (e.g., field strength, scanner, and acquisition parameters) create substantial, non-biological variance in image appearance (Guo et al., 2024).

Image harmonization has emerged as a critical processing step to mitigate domain shift (Hu et al., 2023; Abbasi et al., 2024). Early deep learning approaches, such as CycleGAN (Zhu et al., 2020; Modanwal et al., 2020), learn mappings between two specific domains (e.g., Scanner A-to-B). While effective, these methods are not scalable in the clinical setting as they require a unique model to be trained for every pair of domains. Other approaches adopt arbitrary style-transfer frameworks such as StarGANv2 (Choi et al., 2020), MUNIT (Huang et al., 2018) and Adaptive Instance Normalization (AdaIN) (Huang and Belongie, 2017), which offer greater flexibility than paired domain-to-domain translation. These models explicitly disentangle content from style, allowing translation to be performed by injecting the style extracted from a target image. Extensions of this idea have also been applied to medical imaging (Ouyang et al., 2021; Dewey et al., 2020), where it is shown that content–style (anatomy-contrast) separation can improve robustness to scanner and protocol variability in brain MRI. More advanced architectures, such as HACA3 (Zuo et al., 2023), enhance the disentanglement of anatomy and contrast by explicitly modelling content and style factors, resulting in improved anatomical fidelity and more consistent harmonization across different contrasts. However, these methods share a fundamental limitation: they require a target image to serve as a style example, which is neither always available nor ideal from a utility point-of-view.

Recent advances in vision–language modelling have opened new possibilities for text guided image synthesis and domain adaptation in both natural and medical imaging. Contrastive Language–Image Pre-training (CLIP) (Radford et al., 2021) established a shared embedding space that aligns visual and textual semantics, inspiring text-guided stylization methods in natural images such as CLIP-Styler (Kwon and Ye, 2021) and StyleGAN-NADA (Gal et al., 2021), where text prompts can steer generative models toward new appearance domains without requiring paired supervision. In medical imaging, recent work has shown that CLIP-guided conditioning can use textual metadata to modulate image appearance and enable contrast-aware synthesis (Wang et al., 2025a,b). Although promising, these approaches are typically trained on curated public datasets that lack the range of contrasts, anatomies, and scanner vendors found in real clinical environments. As a result, their ability to serve as robust, general-purpose harmonization or contrast-normalization frameworks remains limited. Addressing this gap, MR-CLIP (Avci et al., 2025b,a) introduced and enabled a more scalable alternative by learning contrast representations directly from large-scale, heterogeneous DICOM metadata–image pairs sourced from real-world clinical environments, enabling broader contrast understanding across varied acquisition settings.

Building on these foundations, we propose **DIST-CLIP** (**DI**sentangled **S**tyle **T**ransfer with **CLIP** Guidance), a unified framework for arbitrary MRI harmonization that can leverage either target images or DICOM metadata as guidance. To ensure robust harmonization across diverse scanners, protocols, and contrasts, our framework explicitly separates

anatomical structure from acquisition-dependent contrast. This separation creates a clean anatomical representation that can be accurately modulated with style information encoded via MR-CLIP embeddings through a novel Adaptive Style Transfer (AST) module. By combining bi-modal guidance with disentangled anatomy–contrast representations, DIST-CLIP achieves flexible, accurate, and highly generalizable harmonization across heterogeneous clinical MRI datasets. Our main contributions are as follows:

- We introduce DIST-CLIP, a unified style transfer framework for MRI harmonization that flexibly incorporates guidance from target images or textual metadata by leveraging CLIP's robust joint embedding space.

- We propose a novel Adaptive Style Transfer (AST) module that injects contrast/style information from CLIP embeddings into disentangled anatomical content for precise and controllable harmonization.

- DIST-CLIP demonstrates robust performance across diverse clinical imaging settings, maintaining high zero-shot generalization capabilities when evaluated on an external, out-of-distribution public dataset.

## 2. Methods and Materials

The core objective of DIST-CLIP is to translate a source MRI scan into a target domain, defined either by a target image or by acquisition metadata, while strictly preserving the underlying anatomical structure. To achieve this, the framework explicitly disentangles each scan into two complementary representations: anatomical content and image contrast. As illustrated in Figure 1A, the Anatomy Mapper extracts the structural content of the scan, while the CLIP encoders captures contrast information from either images or metadata. These disentangled representations are then fused in the Style Fusion Decoder (SFD), which synthesizes the final image by integrating the anatomical structure with the desired contrast.

**Anatomy Mapper**  The first stage aims to isolate the underlying anatomical structure from imaging physics related variations. The source image $I_{src}$ is projected by the Anatomy Mapper into a structural representation $\beta_{src}$, which captures the brain's morphological geometry, such as tissue boundaries and global anatomical layout, while remaining largely invariant to intensity and contrast differences. Importantly, because different MRI sequences emphasize different tissue properties, the mapper is designed to preserve subtle biologically meaningful variations rather than enforcing strict cross-contrast uniformity. The resulting representation $\beta_{src}$ provides a contrast-agnostic structural foundation for the SFD, onto which the style module injects the desired contrast embedding $\theta$.

**Style Encoding via CLIP**  We leverage pre-trained MR-CLIP encoders to model image contrast, which project both images and DICOM metadata into a shared contrast-aware embedding space. This allows DIST-CLIP to accept two forms of guidance: a target image, processed by the image encoder ($E_I$) to produce a visual style embedding $\theta_i$, or the corresponding metadata, processed by the metadata encoder ($E_M$) to produce a textual style embedding $\theta_m$. To ensure the model interprets both modalities consistently, at each training iteration, we randomly condition the decoder on $\theta_i$ or $\theta_m$ (with $p = 0.5$). Thus, at

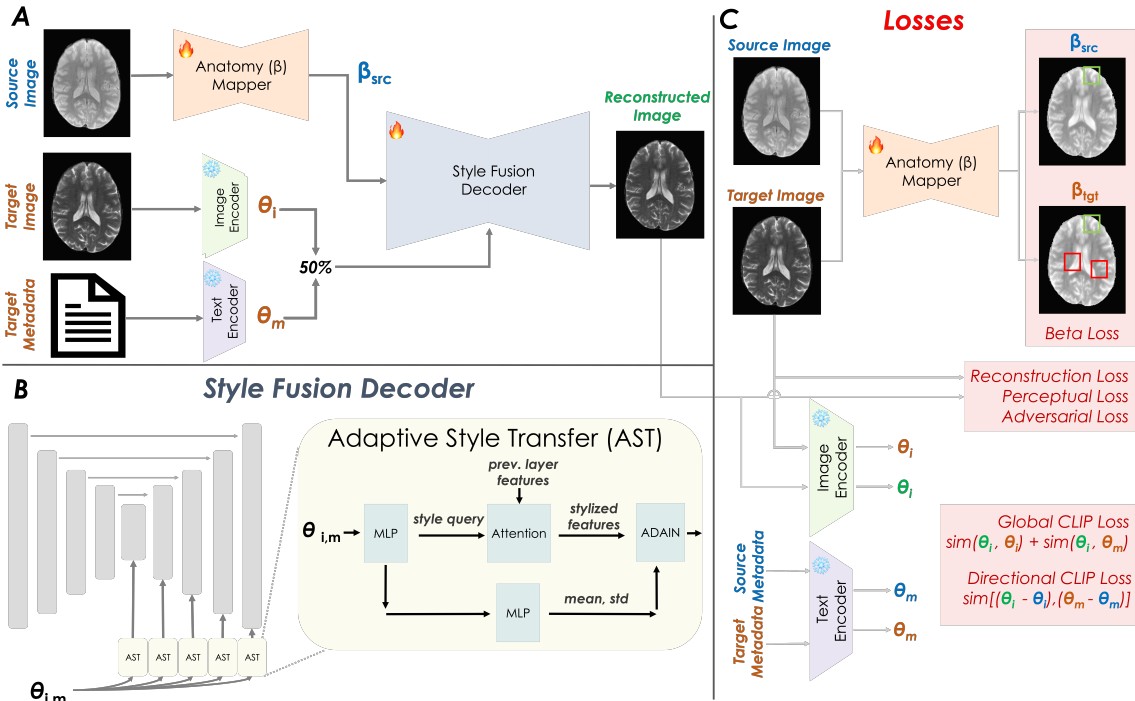

Figure 1: **Overview of the DIST-CLIP framework.** (**A**) Overall architecture: source image is processed by the *Anatomy Mapper* to extract a disentangled anatomical representation ($\beta_s$). In parallel, a style embedding ($\theta_i$ or $\theta_m$) is derived from either a target image or metadata using pre-trained CLIP encoders. The *Style Fusion Decoder (SFD)* integrates these anatomy and style representations to synthesize the final image with the desired appearance. (**B**) Detailed structure of the SFD, which adaptively fuses anatomical and style features through *Adaptive Style Transfer (AST)* blocks. (**C**) Loss suite used for training, enforcing anatomical preservation, reconstruction fidelity and style consistency.

inference time, DIST-CLIP can be guided by either an image or metadata alone, since $\theta_i$ and $\theta_m$ were pre-trained to be aligned with each other.

**Style Fusion and Image Synthesis** The final harmonization is performed by the SFD, which synthesizes the output image by fusing the target style embedding $\theta$ with the anatomical representation $\beta_{src}$. As shown in Figure 1B, the decoder incorporates the desired contrast by injecting $\theta$ at every decoding layer through an Adaptive Style Transfer (AST) module. Within AST, an MLP maps the style embedding to a query vector that drives an attention mechanism, allowing the model to selectively modulate the previous layer's features and capture localized style–content interactions. The resulting attention output is applied to a spatial AdaIN layer, performing final feature modulation to refine the reconstruction. This progressive, layer-wise modulation allows the synthesized volume to accurately reflect the target contrast while strictly preserving the underlying anatomical structure.

## 2.1. Loss Functions

As shown in Figure 1C, DIST-CLIP is trained with a composite objective that jointly enforces anatomical preservation, high-fidelity reconstruction, and CLIP-guided style consistency, with the specific loss components described below.

To ensure that the anatomical representations ($\beta$) preserve the fine-grained biological structure and simultaneously enforces contrast-invariance, we adopt a contrastive patch-alignment strategy from HACA3 (Zuo et al., 2023). Different MRI sequences highlight different tissue signals, so enforcing strict feature equality across contrasts is overly restrictive. Instead, this loss works locally, aligning patches at corresponding locations in the source ($\beta_{src}$) and target ($\beta_{tgt}$) maps, while simultaneously pushing apart patches that do not correspond. This emphasis on relative similarity ensures structural fidelity while tolerating necessary feature space variations between different image contrasts. Formally, the loss is defined as a patch-wise InfoNCE (van den Oord et al., 2019) objective:

$$\mathcal{L}_\beta = \mathbb{E}_{I_{src}, I_{tgt}} \sum_{i \in P} \left[ -\log \frac{e^{\mathrm{sim}(z_i, z_i')/\tau}}{\sum_{k \in P} e^{\mathrm{sim}(z_i, z_k')/\tau}} \right]$$

where $P$ is the set of all patch locations. For each source patch $z_i$ in $\beta_{src}$, $z_i'$ is the corresponding patch in $\beta_{tgt}$, while $z_k'$ (for all $k \in P$) represents *all* patches in the target map, $\mathrm{sim}(\cdot, \cdot)$ is the cosine similarity, and $\tau$ is the temperature parameter.

Additionally, to encourage the harmonized outputs to retain realistic textures and fine details, we optimize three complementary losses. A pixel-wise $L_1$ reconstruction loss ($\mathcal{L}_{rec}$) preserves low-level structural similarity, a VGG-based perceptual loss ($\mathcal{L}_{perc}$) encourages high-level semantic and textural fidelity, and an adversarial loss ($\mathcal{L}_{adv}$) with a patch discriminator promotes photorealistic appearance:

$$\mathcal{L}_{rec} = \mathbb{E}\left[\|I_{tgt} - I_{rec}\|\right]$$
$$\mathcal{L}_{perc} = \mathbb{E}\left[\sum_l \|\phi_l(I_{tgt}) - \phi_l(I_{rec})\|\right] \tag{1}$$
$$\mathcal{L}_{adv} = \mathbb{E}_{I_{tgt} \sim p_{real}}[\log D(I_{tgt})] + \mathbb{E}_{I_{rec} \sim p_g}[\log(1 - D(I_{rec}))]$$

where $\phi_l$ represents the feature maps extracted from the $l$-th layer of a pre-trained VGG-19 network, and $D$ denotes the discriminator.

Style accuracy is enforced through losses computed in the joint MR-CLIP embedding space, allowing alignment to image and metadata based contrast targets. The global CLIP loss encourages the synthesized image to match the target contrast by maximizing similarity between the output embedding $E_I(I_{rec})$ and the target embeddings $\theta_i$ and $\theta_m$:

$$\mathcal{L}_{\mathrm{global}} = \frac{1}{2}\big(1 - \mathrm{sim}(E_I(I_{\mathrm{rec}}), \theta_i)\big) + \frac{1}{2}\big(1 - \mathrm{sim}(E_I(I_{\mathrm{rec}}), \theta_m)\big) \tag{2}$$

To ensure precise and stable contrast transfer, we incorporate the directional CLIP loss from StyleGAN-NADA (Gal et al., 2021), which explicitly enforces that the direction of change in the image embedding matches the direction of change in the metadata embedding:

$$\mathcal{L}_{dir} = 1 - \frac{\Delta I \cdot \Delta M}{\|\Delta I\|\|\Delta M\|} \tag{3}$$

where $\Delta I = E_I(I_{rec}) - E_I(I_{src})$ and $\Delta M = E_M(M_{tgt}) - E_M(M_{src})$.

The final objective function is a weighted sum of all components:

$$\mathcal{L}_{total} = \lambda_\beta \mathcal{L}_\beta + \lambda_{rec}\mathcal{L}_{rec} + \lambda_{perc}\mathcal{L}_{perc} + \lambda_{adv}\mathcal{L}_{adv} + \lambda_{global}\mathcal{L}_{global} + \lambda_{dir}\mathcal{L}_{dir} \qquad (4)$$

where the $\lambda$ hyperparameters control the relative importance of each term.

### 2.2. Implementation Details and Evaluation Protocol

DIST-CLIP is implemented in PyTorch. The Anatomy Mapper is realized as a U-Net, with Instance Normalization applied in the first layer to extract contrast-invariant anatomical representations. The SFD is also a U-Net with double the channel width of the Anatomy Mapper, allowing richer feature modulation. AST modules are inserted at the bottleneck and upsampling layers to inject 512-dimensional MR-CLIP embeddings into the feature maps. To control computational cost, 8-head multi-head attention block is used at the bottleneck, while lighter linear-attention blocks were employed in the upsampling layers. Training is performed using the Adam optimizer (Kingma and Ba, 2017) (learning rate $1.0 \times 10^{-4}$, batch size 25) for 15 epochs. The loss weights were tuned such that all loss terms lie within a similar magnitude range, ensuring balanced contributions and preventing any single objective from dominating the optimization: $\lambda_\beta = 0.1$, $\lambda_{rec} = 10.0$, $\lambda_{perc} = \lambda_{adv} = 1.0$, and $\lambda_{global} = \lambda_{dir} = 1.0$.

To evaluate our approach, we compare DIST-CLIP against TUMSyn and HACA3 using PSNR and SSIM computed within the brain region to ensure fair comparison across methods. We follow the original HACA3 and TUMSyn implementations, and use the released pre-trained weights. For HACA3, we apply the model to images that include the skull. For quantitative evaluation, metrics are computed only within the brain mask, while for visualization we additionally apply brain masking to highlight relevant anatomical structures.

### 2.3. Dataset and Preprocessing

We utilize a large-scale dataset of brain MRI scans acquired from King's College Hospital and Guy's and St Thomas' NHS Foundation Trust. The dataset comprises 21,115 volumes from 8,466 subjects across 8,820 studies. The data are split into training, validation, and test sets at the patient level to ensure no subject overlap between splits. The number of cross-contrast source–target pairs included in the test set is detailed in Table 2 in the Appendix. All volumes are rigidly registered to the MNI atlas with 1.0 $mm^3$ resolution, followed by skull-stripping using SynthStrip (Hoopes et al., 2022). For training, every second slice from the central 100 slices is used, providing sufficient anatomical context for the whole brain while keeping the data size manageable whereas all slices are used for testing. We construct metadata prompts as shown in Appendix Table 3 which follows methodology of MR-CLIP, utilizing a comprehensive set of DICOM fields: *Echo Time, Repetition Time, Inversion Time, Manufacturer, Scanner Model, Imaging Plane, Field Strength, Sequence Type, Sequence Variant, Series Description, Flip Angle.* The slicing plane (axial, coronal, or sagittal) is determined based on voxel resolution, prioritizing the highest-resolution dimension; isotropic images default to the axial plane. Input images are resized to $224 \times 224$ and normalized to $[0, 1]$.

## 3. Results

We evaluate DIST-CLIP across multiple contrast translation tasks, comparing it against state-of-the-art baselines TUMSyn and HACA3 using image quality metrics PSNR and SSIM to assess reconstruction fidelity, alongside qualitative visual results. Beyond reconstruction quality, we further analyze the learned anatomical representations, evaluate generalization on out-of-distribution data, and conduct an ablation study to quantify the contribution of key components.

The quantitative results in Figure 2 summarize harmonization performance across the four major MRI contrasts (FLAIR, T1w, T2w, PDw), with PSNR shown in the top row and SSIM in the bottom row. Each heatmap reflects available pairwise translation tasks, where rows denote the source contrast and columns denote the target contrast. DIST-CLIP achieves consistently strong reconstruction quality, with average PSNR values reaching up to 31.9 dB and SSIM scores up to 0.973 (PDw→T2w). These high ranges demonstrate its robust perceptual and structural fidelity across tasks, outperforming all baselines by a clear margin. Importantly, although images offer richer and more precise visual cues than textual metadata, the metadata-guided model (DIST-CLIP/T) still achieves performance that is only marginally lower than image-guided model (DIST-CLIP/I). This small gap underscores that, despite the noise and variability inherent in clinical metadata, DICOM text remains a highly effective semantic descriptor of contrast. As a result, DIST-CLIP can deliver robust, high-quality harmonization even when no target images are available at inference time.

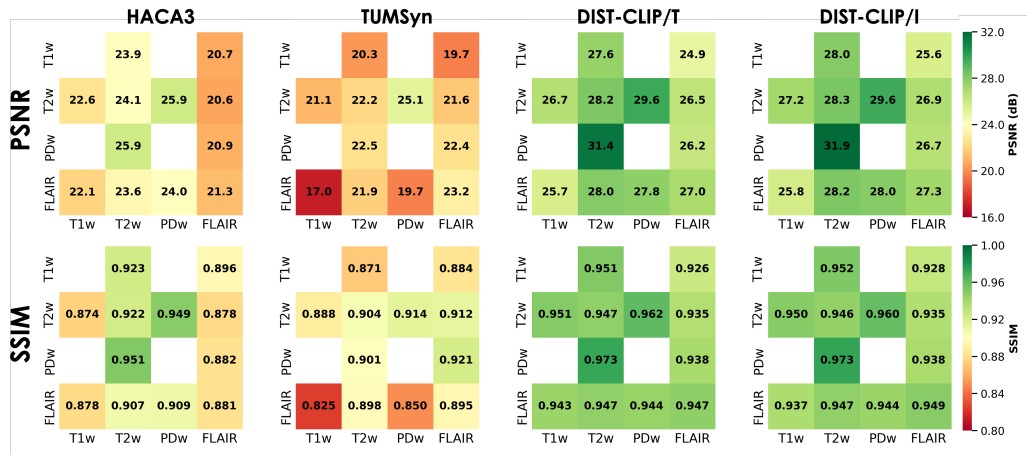

Figure 2: Quantitative evaluation of cross-contrast harmonization. Heatmaps show PSNR (top row) and SSIM (bottom row) for all present bi-directional translation tasks across T1w, T2w, PDw, and FLAIR MRI sequences. Rows represent the source contrast, and columns represent the target contrast, comparing DIST-CLIP (Image/I and Text/T guided) against HACA3 and TUMSyn.

The visual comparison in Figure 3 highlights clear qualitative differences across methods. Analysis of the baseline models reveals specific output characteristics: HACA3 generates visually sharp images but have moderate style fidelity, while TUMSyn generates overly

smoothed results with visible grid artifacts. In contrast, both DIST-CLIP variants (text-guided /T and image-guided /I) produce harmonized images that closely match the target, maintaining sharp anatomical boundaries and realistic tissue contrast. The near-identical appearance of DIST-CLIP/T and DIST-CLIP/I further demonstrates the strength of the MR-CLIP embedding space in aligning DICOM metadata with image-based cues.

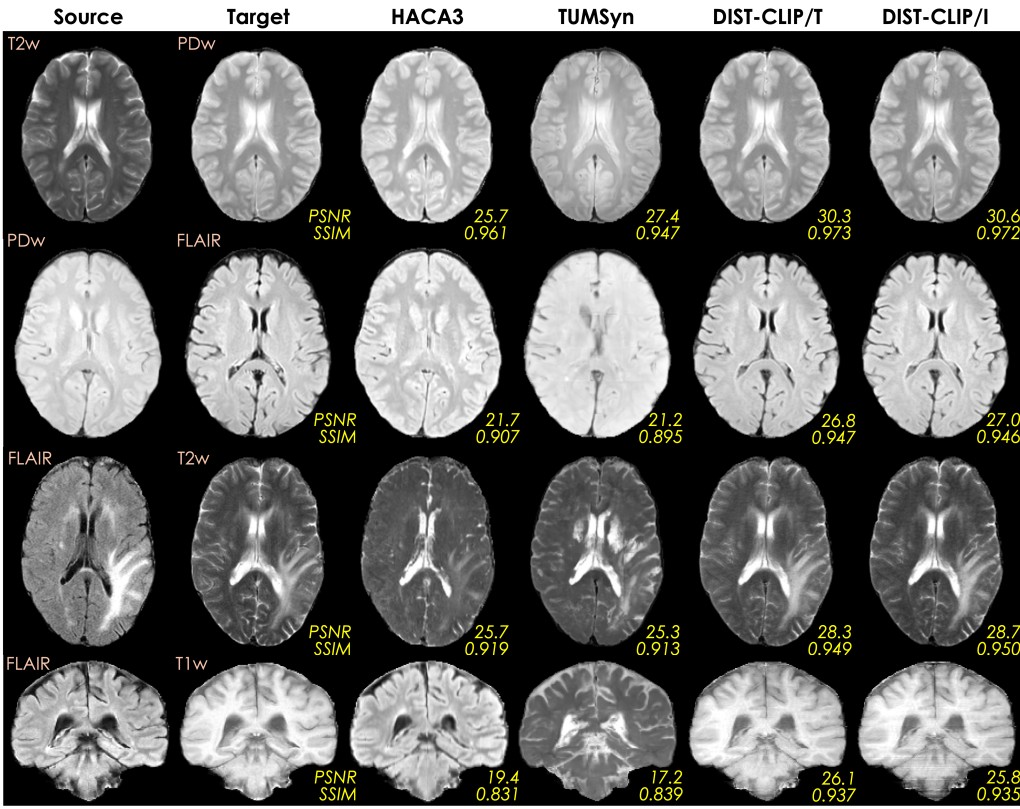

Figure 3: Qualitative assessment of cross-contrast harmonization. Outputs from baselines (HACA3, TUMSyn) are shown alongside the DIST-CLIP framework (text-guided /T and image-guided /I). Inset PSNR (dB) and SSIM scores confirm DIST-CLIP's high structural and visual fidelity.

Figure 4 shows the anatomical maps ($\beta$-maps) extracted by the Anatomy Mapper. Across all input contrasts, these maps preserve the underlying structural geometry while suppressing acquisition-dependent intensity variations, demonstrating effective disentanglement of anatomy from contrast and provides a robust backbone for style injection. Although the $\beta$-maps appear visually similar across sequences, subtle anatomical differences remain visible in the zoomed regions, reflecting genuine variations in tissue signal across contrasts. This behaviour validates the use of a contrastive alignment loss, ensuring that the mapper preserves fine structural information while discarding contrast-specific appearance.

Furthermore, we investigate the contribution of DIST-CLIP components through an ablation study summarized in Table 1. Replacing the AST blocks with a standard AdaIN

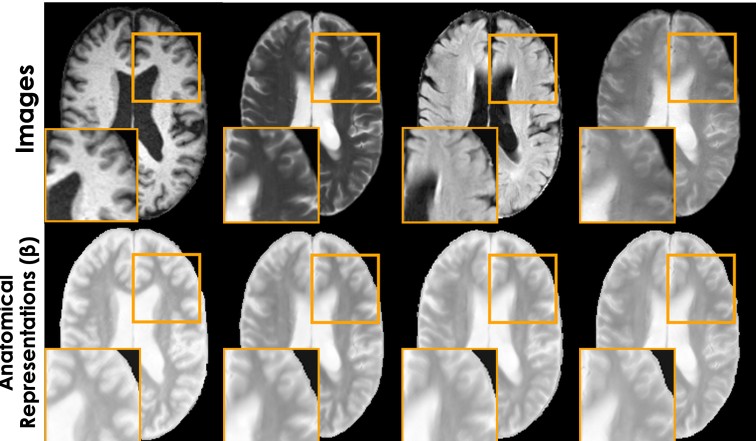

Figure 4: Anatomical Representations. The top row displays the source MR Images (T1w, T2w, FLAIR, T2*w, respectively), and the bottom row shows their corresponding contrast-invariant anatomical ($\beta$) representations.

layer results in a significant performance drop, indicating that attention-driven modulation is crucial for precise contrast transfer. We also evaluate the setting without anatomical disentanglement ("no $\beta$ disentanglement"), where the source image is directly fed into the Style Feature Decoder (SFD) without passing through the Anatomy Mapper. This leads to reduced PSNR and SSIM, confirming that the Anatomy Mapper plays an important role in separating anatomical structure from contrast information rather than simply acting as an additional processing block. We further analyze the contribution of different guidance modalities. In the only metadata guidance and only image guidance settings, the corresponding modality is not provided as an explicit style input to the decoder, but it is still incorporated through the CLIP-based alignment loss during training. As a result, the model can still benefit from weak supervisory signals from the unused modality, which explains the relatively small performance differences across these variants. Overall, the full DIST-CLIP model, which jointly leverages both image-based and metadata-based guidance at the architectural level and through CLIP supervision, achieves the most balanced and consistent performance across PSNR and SSIM, demonstrating that complementary multi-modal guidance improves robust harmonization.

External evaluation on the OASIS-3 dataset (Figure 5) highlights the strong zero-shot generalization capabilities of DIST-CLIP. As shown in Figure 5A, both the image-guided and text-guided models generate synthesized images with sharp anatomical details, even though these combinations of acquisition parameters and image types were never seen during training. Quantitative results in Figure 5B further demonstrate that DIST-CLIP/I matches or exceeds the performance of TUMSyn, which was trained directly on OASIS-3, underscoring the robustness of our disentangled architecture. While DIST-CLIP/T experiences some challenges in style transfer with unseen metadata, it maintains strong anatomical fidelity, as indicated by high SSIM scores. Taken together, bi-modal guidance allows DIST-CLIP to achieve zero-shot harmonization across cohorts, despite variations in metadata.

Table 1: Ablation study of DIST-CLIP components showing image and text guided performance, averaged over all contrast translation tasks.

| | PSNR ↑ | | SSIM ↑ | |
|---|---|---|---|---|
| **Model Variant** | **Image** | **Text** | **Image** | **Text** |
| AdaIN instead of AST blocks | 21.6 | 21.9 | 0.889 | 0.900 |
| No $\beta$ disentanglement (no Anatomy Mapper) | 28.5 | 28.2 | 0.949 | 0.950 |
| Trained with only metadata guidance | 27.6 | 28.3 | 0.941 | 0.950 |
| Trained with only image guidance | **29.0** | 27.7 | **0.953** | 0.949 |
| **Full DIST-CLIP** | 28.7 | **28.4** | 0.951 | **0.951** |

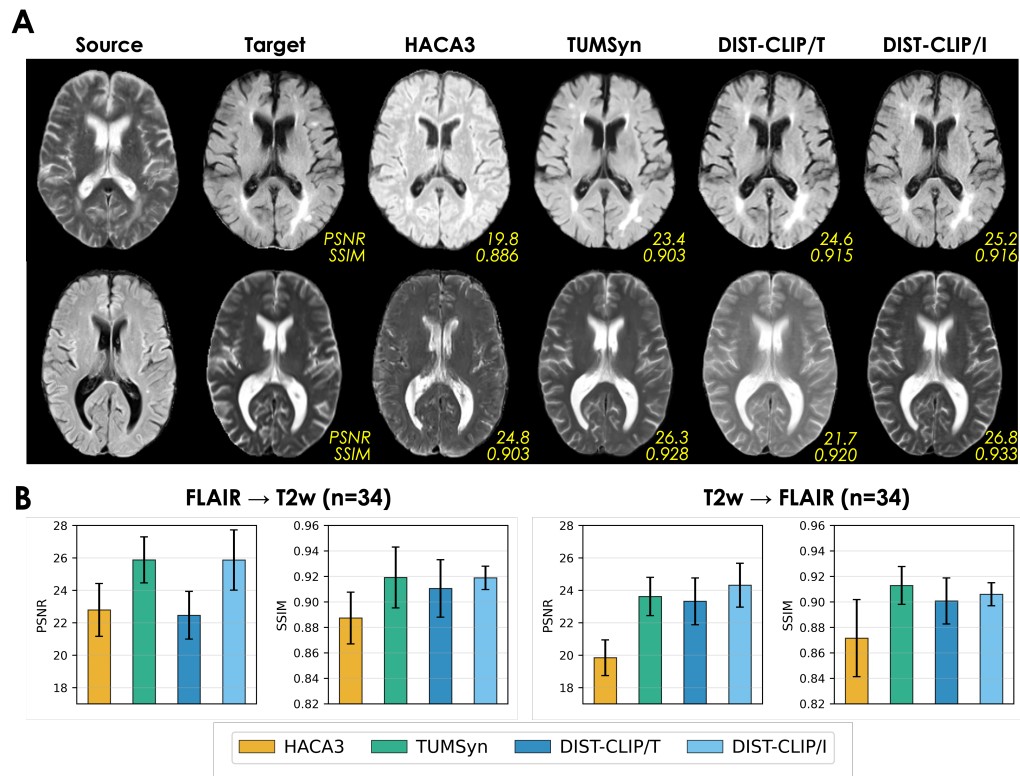

Figure 5: Qualitative and quantitative results on the OOD (OASIS-3) dataset. (**A**) Visual comparison of harmonization performances. (**B**) Quantitative analysis of bidirectional translation ($n = 34$) measured by PSNR and SSIM, demonstrating that DIST-CLIP performs on par with or better than state-of-the-art methods, even without being trained on this dataset.

## 4. Discussion

In this work, we introduced DIST-CLIP, a unified framework for medical image harmonization that seamlessly supports both image-guided and text-guided synthesis. By leveraging the joint embedding space of pre-trained MR-CLIP encoders, DIST-CLIP disentangles anatomical structure from acquisition-specific contrast and enables precise modulation of style using either target images or DICOM metadata. Central to this flexibility is the AST module, which enables fine-grained contrast injection while preserving anatomical fidelity. Across extensive evaluations on both large-scale clinical data and the external OASIS-3 cohort, DIST-CLIP consistently outperforms unimodal baselines and exhibits strong generalization, including in zero-shot settings. While promising, our approach currently operates on 2D slices to leverage powerful pre-trained 2D encoders. This design provides computational efficiency but does not explicitly enforce continuity along the through-plane, indicating a natural opportunity for future extensions to fully 3D representations to improve volumetric coherence. Moreover, although our framework leverages DICOM metadata for contrast control, real-world clinical metadata may contain inconsistencies or missing fields. A more systematic evaluation of how such metadata variability affects harmonization performance would further strengthen the robustness of the framework in practical clinical settings. Additionally, although harmonized outputs achieve high visual and quantitative fidelity, it remains important to evaluate their downstream utility in tasks such as tissue segmentation or morphometric quantification to ensure that subtle but clinically meaningful intensity boundaries are preserved. Overall, DIST-CLIP advances the goal of robust, scalable MRI harmonization by enabling high-quality contrast-controlled synthesis, paving the way for more consistent multi-site imaging pipelines and more reliable deployment of clinical AI models.

## Acknowledgments

This work was supported by the UK Engineering and Physical Sciences Research Council (EPSRC) [Grant reference number EP/Y035216/1] Centre for Doctoral Training in Data-Driven Health (DRIVE-Health) at King's College London, with additional support from deepc GMBH. Further support is given by Scientific and Technological Research Council of Türkiye (TÜBİTAK) under 2213-A Overseas Graduate Scholarship.

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

## Appendix A. Supplementary Dataset Information

|        | T1w | T2w | PDw | FLAIR |
|--------|-----|-----|-----|-------|
| T1w    | -   | 32  | -   | 11    |
| T2w    | 32  | 34  | 136 | 185   |
| PDw    | -   | 136 | -   | 22    |
| FLAIR  | 11  | 185 | 22  | 4     |

Table 2: Number of available cross-contrast source–target pairs included in the test set, representing translations evaluated by the model. Rows represent the source contrast, and columns represent the target contrast. T1w→T1w and T1w→PDw are not available in the test set.

Table 3: Exemplar common metadata descriptions.

A brain MRI, *plane* axial, **Scanner** (**Manufacturer**, **Model**, **Field Strength**): (GE, Signa_HDxt, 1.5), **Acquisition** (**Description**, **Sequence**, **Variant**): (Ax T2 FLAIR, SE_IR, SK), **Imaging Parameters** (**Echo Time**, **Repetition Time**, **Inversion Time**, **Flip Angle**): (0.129, 8.002, 2, 90)

A brain MRI, *plane* axial, **Scanner** (**Manufacturer**, **Model**, **Field Strength**): (GE, Signa_HDxt, 1.5), **Acquisition** (**Description**, **Sequence**, **Variant**): (Ax PD/T2, SE, SK), **Imaging Parameters** (**Echo Time**, **Repetition Time**, **Inversion Time**, **Flip Angle**): (0.0203, 5.9, NONE, 90)

A brain MRI, *plane* axial, **Scanner** (**Manufacturer**, **Model**, **Field Strength**): (Siemens, Aera, 1.5), **Acquisition** (**Description**, **Sequence**, **Variant**): (t2_tse_tra_384_p2, SE, SK_SP_OSP), **Imaging Parameters** (**Echo Time**, **Repetition Time**, **Inversion Time**, **Flip Angle**): (0.087, 3.96, NONE, 150)

A brain MRI, *plane* axial, **Scanner** (**Manufacturer**, **Model**, **Field Strength**): (Siemens, Avanto, 1.5), **Acquisition** (**Description**, **Sequence**, **Variant**): (pd+t2_tse_tra, SE, SK_SP_OSP), **Imaging Parameters** (**Echo Time**, **Repetition Time**, **Inversion Time**, **Flip Angle**): (0.014, 3.68, NONE, 150)

A brain MRI, *plane* coronal, **Scanner** (**Manufacturer**, **Model**, **Field Strength**): (GE, SIGNA_HDx, 1.5), **Acquisition** (**Description**, **Sequence**, **Variant**): (Cor T1, SE, NONE), **Imaging Parameters** (**Echo Time**, **Repetition Time**, **Inversion Time**, **Flip Angle**): (0.02, 0.4, NONE, 90)

