# OpenReview forum: "DIST-CLIP: Arbitrary Metadata and Image Guided MRI Harmonization via Disentangled Anatomy-Contrast Representations"
_MIDL.io/2026/Conference — MIDL 2026 Poster_

### Official Review · Reviewer_QwJv · 2026-01-06

**Confidence:** 4
**Preliminary Rating:** 4
**Final Rating:** 4

**Summary:**

This paper proposes a novel CLIP-based disentangled framework to guide MRI harmonization. The framework adopts a dual-branch input design. Information from both the source image and the target image is extracted and fed into the SFD module to support subsequent image reconstruction. Experimental results demonstrate that this approach outperforms existing methods like HACA3 and TUMSyn.

**Strengths:**

1. The method supports both image-guided and metadata-guided MRI harmonization within a unified framework.

2. It uses real DICOM metadata to guide harmonization, rather than relying on simplified text descriptions.

3. Experiments on an external dataset further demonstrate the effectiveness and generalization ability of the proposed method.

**Weaknesses:**

1. All experiments are evaluated using image quality metrics such as SSIM and PSNR. It would be beneficial to further
validate the proposed method on downstream tasks to demonstrate its practical impact.

2. Experiments are conducted on brain MRI data. The proposed approach may lack cross-organ generalizability.

**Detailed Comments:**

1. Provide a more detailed explanation of Table 1. Except for the first row, the results across different settings are very close.

2. Explain how the loss weights are selected.

**Justification Of Final Rating:**

The authors solve all my previous questions and also provide detailed explanation for the reason of using brain MRI data. I am happy to accept this paper and I am looking forward to your final paper.
Thanks!

**Justification Of The Preliminary Rating:**

My rating is based on the methodological design (image + meta-data from dicom) and practical relevance of the proposed approach. Also, their experiments especially the external dataset validation are important to my rating.

**Questions To Address In The Rebuttal:**

In Table 1, could you please clarify the setting “No β disentanglement (no Anatomy Mapper)”? It appears that the anatomy mappr does not contribute to the model in this case. Does this setting correspond to directly feeding the source image into the SFD module?

---

> ### Author Response · Authors · 2026-01-23
>
> We thank the reviewer for their insightful feedback and for recognizing the ‘practical relevance’ of DIST-CLIP, particularly our use of real DICOM metadata and external validation. Below, we address the specific points raised.
>
> ### Lack of Downstream Task Evaluation
>
> We agree with the reviewer that downstream task evaluation (e.g., segmentation) would further strengthen the paper. In this submission, we intentionally focus on validating accurate and controllable contrast harmonization, as downstream performance is only meaningful once translation fidelity is established. We are actively working on an extension, and we plan to include such results in an extended version of the paper.
>
> ### Limited Evaluation to Brain MRI
>
> We agree that the current experimental evaluation is limited to brain MRI data. This choice was made deliberately to control for anatomical variability and to focus the study on the proposed disentanglement and multi-modal guidance mechanisms, rather than on cross-anatomy generalization. However, the DIST-CLIP architecture is fundamentally organ-agnostic. The Anatomy Mapper and Style Feature Decoder do not rely on brain-specific priors, atlases, or task-specific labels. The underlying mechanism of separating structural geometry from acquisition style is applicable to other regions or modalities. We will explicitly clarify this architectural flexibility and our plans for broader evaluation in the revised manuscript.
>
> Regarding the comments on Table 1, we acknowledge that the explanations in the current submission could be clearer, and we appreciate the opportunity to clarify them which we will incorporate to the revised manuscript:
>
> - **No β disentanglement (no Anatomy Mapper)** refers to the setting where the source image is directly fed into the Style Feature Decoder (SFD), without anatomical disentanglement. This confirms the role of the Anatomy Mapper in separating structure from style rather than simply acting as an additional processing block.
> - **Only metadata guidance** and **only image guidance** indicate that the corresponding modality is *not* used as an explicit style input to the decoder, but it is still incorporated through the CLIP-based alignment loss during training. As a result, the model can still learn weak supervisory signals from the unused modality, which explains why the performance remains relatively close across these variants.
> - Importantly, the **full DIST-CLIP framework**, which jointly leverages image-based and metadata-based guidance both architecturally and through CLIP supervision, achieves the most balanced and consistent performance across PSNR and SSIM. This supports our core claim that complementary multi-modal guidance is beneficial beyond using either modality alone.
>
> ### Loss Weighting
>
> Loss weights are set to contribute equally to ensure that no single loss term dominates the optimization. We agree that this detail was not clearly stated in the paper, and we will make this explicit in the revised manuscript.

---

### Official Review · Reviewer_XB8Y · 2026-01-08

**Confidence:** 4
**Preliminary Rating:** 4

**Summary:**

This manuscript presents DIST-CLIP, a novel MRI harmonization framework that separates anatomical structure from acquisition-specific contrast using pre-trained CLIP encoders, guided by either target images or DICOM metadata. An Adaptive Style Transfer (AST) module enables precise contrast transfer while preserving anatomical fidelity. Evaluated on large clinical datasets and the OASIS-3 cohort, DIST-CLIP outperforms HACA3 and TUMSyn in PSNR, SSIM, visual realism, and zero-shot generalization, offering a unified, metadata-aware solution for scalable multi-site medical AI.

**Strengths:**

The paper’s main strength is DIST-CLIP’s unified and flexible harmonization framework, which supports guidance from either target images or DICOM metadata, overcoming a key limitation of image-only methods and improving clinical applicability. Its innovative use of CLIP (MR-CLIP) to jointly encode images and acquisition metadata enables robust, scalable modeling of real-world MRI heterogeneity. The explicit disentanglement of anatomy and contrast, together with the novel Adaptive Style Transfer (AST) module, allows precise style injection while preserving structural fidelity.
The method is rigorously evaluated on large clinical datasets and the OASIS-3 cohort, with strong quantitative, qualitative, ablation, and zero-shot generalization results. Overall, the work shows good novelty and scientific rigor and significant potential impact, offering a practical and scalable solution for multi-site MRI harmonization and reliable clinical AI deployment.

**Weaknesses:**

Despite its strengths, the paper has several limitations.
The robustness of metadata-based guidance is not thoroughly validated under realistic noisy or incomplete metadata scenarios.
The 2D slice-based processing limits volumetric coherence, with no analysis of through-plane consistency in 3D volumes.
Evaluation relies mainly on low-level metrics (PSNR/SSIM), lacking validation of downstream tasks to demonstrate clinical relevance.
The disentanglement quality is only indirectly assessed, without explicit measures of anatomical contrast invariance.
Finally, training, testing data overlap, and baseline selection are not clearly justified. Addressing these issues would strengthen the evidence for real-world clinical applicability.

**Detailed Comments:**

Figure/Table consistency: Ensure correct referencing of Figures 1A–1C and Tables 2–3; add a brief pointer to appendix tables in the main text.
Loss notation clarity: Clarify the adversarial loss in Eq. (1); the expectation term appears inconsistent and may confuse readers.
Metadata prompt illustration: Add one concise metadata-to-text prompt example in the Methods section for clarity.
Acronym definition: Define AST (Adaptive Style Transfer) at its first full mention in the main text.
Baseline transparency: Explicitly state that DIST-CLIP was not fine-tuned on OASIS-3 for zero-shot evaluation.
Broader impact: Consider a brief ethical or clinical-use discussion on harmonized/synthesized data.
Typos & formatting:
Standardize “style transfer” capitalization in keywords.
Fix missing bracket in Eq. (1).
Simplify phrasing to “an 8-head attention block.”
2D limitation clarification: Briefly note potential through-plane inconsistencies to better motivate future 3D extensions.

**Justification Of The Preliminary Rating:**

This paper receives a Weak Accept for proposing a novel, properly designed, and practical MRI harmonization framework that includes the field by unifying image and metadata guidance through a disentangled CLIP-based architecture. The methodology is sound, ablations are appropriate, and the strong zero-shot generalization on OASIS-3 supports the quality of the learned representations.
The rating remains weak due to limited validation depth. Evaluation focuses on image fidelity and visual quality (PSNR/SSIM), without downstream task experiments to confirm preservation of clinically meaningful features.

**Questions To Address In The Rebuttal:**

Further Comments & Minor Suggestions:
Figure/Table consistency: Ensure correct referencing of Figures 1A–1C and Tables 2–3; add a brief pointer to appendix tables in the main text.
Loss notation clarity: Clarify the adversarial loss in Eq. (1); the expectation term appears inconsistent and may confuse readers.
Metadata prompt illustration: Add one concise metadata-to-text prompt example in the Methods section for clarity.
Acronym definition: Define AST (Adaptive Style Transfer) at its first full mention in the main text.
Baseline transparency: Explicitly state that DIST-CLIP was not fine-tuned on OASIS-3 for zero-shot evaluation.
Broader impact: Consider a brief ethical or clinical-use discussion on harmonized/synthesized data.
Typos & formatting:
Standardize “style transfer” capitalization in keywords.
Fix missing bracket in Eq. (1).
Simplify phrasing to “an 8-head attention block.”
2D limitation clarification: Briefly note potential through-plane inconsistencies to better motivate future 3D extensions.

---

> ### Author Response · Authors · 2026-01-23
>
> We thank the reviewer for the thorough evaluation and constructive feedback. We are encouraged by the positive remarks highlighting our work with ‘good novelty and scientific rigor and significant potential impact’, ‘sound methodology’ with ‘strong quantitative, qualitative, ablation, and zero-shot generalization results’ and ‘significant potential impact’. We also appreciate the recognition of the ‘innovative’ use of MR-CLIP for joint image and metadata encoding, the explicit disentanglement of anatomy and contrast. We are grateful for the detailed assessment of our method’s rigor and clinical relevance.
>
> We address the reviewer’s comments and concerns point by point below.
>
> ### Inconsistent and Nosiy Metadata Guidance
>
> We thank the reviewer for raising this point and agree that validating metadata-based guidance under noisy or incomplete scenarios is important. During the training of MR-CLIP, we use a large, uncurated clinical dataset from real hospital data, which naturally contains noise and inconsistencies. The CLIP model is able to learn robust representations despite these imperfections and detect problematic (missing or wrong) metadata, as demonstrated in our previous work ([arXiv:2511.00681](https://arxiv.org/pdf/2511.00681)). Nevertheless, we agree that systematic evaluation of harmonization under realistic noisy and incomplete metadata scenarios is valuable, and we plan to include such analysis in an extended version of the paper.
>
> ### 2D Processing and Volumetric Coherence
>
> We acknowledge that our current 2D slice-based processing may limit volumetric coherence. To qualitatively assess this effect, we include in Fig. 3 (last row) a coronal view of a volume that was processed slice-wise in the axial plane and then visualized in the coronal plane. This provides qualitative insight into the degree of cross-slice consistency achieved under the 2D setting. We will clarify this evaluation protocol in the revised manuscript. Extending DIST-CLIP to full 3D processing is an important future direction; notably, the proposed framework is architecturally agnostic and naturally supports 3D models, and we are actively working on it. We will also incorporate a part to better motivate for extending the work to 3D.
>
> ### Downstream  Evaluation
>
> We agree with the reviewer that evaluating the impact of DIST-CLIP on downstream tasks is important for demonstrating clinical relevance. In this submission, we focus on validating reliable and controllable contrast harmonization as a prerequisite for meaningful downstream analysis. We are actively conducting downstream evaluations and will include these results in the extended version of the paper.
>
> ### Training/test Split and Baseline Belection
>
> We ensure a rigorous separation between training and test sets, with no overlap between subjects. In addition, we evaluate DIST-CLIP on an unseen external dataset (OASIS-3) to demonstrate generalization. Baselines were selected among the state-of-the-art harmonization models that are applicable to arbitrary harmonization tasks, using either image- or metadata-based guidance, to ensure fair and relevant comparisons. We will make these clearer in revised version.
>
> ### Detailed Comments
>
> We sincerely thank the reviewer for the meticulous feedback and detailed comments. We will try to address all of your detailed comments and minor suggestions in the revised manuscript.

---

### Official Review · Reviewer_qbzY · 2026-01-10

**Confidence:** 5
**Preliminary Rating:** 3
**Final Rating:** 4

**Summary:**

The work introduces DIST-CLIP, a unified framework for medical image harmonization that supports both image-guided and text-guided synthesis by leveraging a joint embedding space of pre-trained MR-CLIP encoders. DIST-CLIP disentangles anatomical structure from acquisition-specific contrast, enabling precise modulation of image style using target images or DICOM metadata. Its central Adaptive Style Transfer (AST) module allows fine-grained contrast injection while maintaining anatomical fidelity throughout synthesis. Evaluations on large-scale clinical datasets and the external OASIS-3 cohort show DIST-CLIP performs well.

**Strengths:**

This paper presents a novel framework. It synthesizes images with respect for anatomy using anatomy-contrast disentanglement. It demonstrates good performance on external datasets and outperforms comparison methods.

**Weaknesses:**

The current implementation operates only on 2D images rather than full 3D volumes. An entire volume is usually needed for downstream tasks like segmentation. How can the 2D results be combined?
The data is skull-stripped, which eliminates some downstream tasks that researchers may be interested. Also, the HACA3 paper shows the model was trained on data involving the skull. Was this method retrained without the skull for comparison?
Although quantitative image metrics (PSNR, SSIM) are strong, the paper does not extensively evaluate the impact of harmonization on key downstream tasks (e.g., segmentation).

**Detailed Comments:**

Please explain how to combine 2D processing results into a 3D volume? Which orientation of slices are used?

**Justification Of Final Rating:**

The authors addressed my concerns and questions in the rebuttal adequately. This resulted in my rating changing from a '3' to a '4'. I think this paper would be of interest to the conference audience and spark great conversion.

**Justification Of The Preliminary Rating:**

The paper has notable strengths and a few unresolved concerns that are mentioned above. The framework is innovative and shows impressive results, but there are concerns that need to be addressed for a more compelling story and trust in the results.

**Questions To Address In The Rebuttal:**

Clarify whether the approach can be adapted for non–skull-stripped or raw clinical MRI scans, and discuss any anticipated performance or generalization issues.
Please clarify how HACA3 was tested since the paper shows it was trained on non-skull-stripped images.
Discuss whether the framework can be extended to use multiple image contrasts or modalities simultaneously for harmonization.
Please clarify why there are missing data points in Fig 3. Why was T1w -> T1w not tested?

---

> ### Author Response · Authors · 2026-01-23
>
> We thank the reviewer for the thorough evaluation of our work and extensive feedback. We appreciate the positive assessment of DIST-CLIP as a ‘novel’ and ‘innovative’ framework, as well as the recognition of its ‘good performance’ and ‘impressive results’ on external datasets and its advantage over existing methods. We also thank the reviewer for the detailed and constructive comments, which we address point by point below.
>
> ### 2D Processing vs 3D Volumes
>
> We agree that many downstream tasks (e.g., segmentation) operate on full 3D volumes. In the current implementation, DIST-CLIP processes 2D slices for computational efficiency to leverage strong 2D CLIP encoders without excessive computational cost. For volumetric reconstruction, 2D slice-wise predictions are stacked back into 3D volumes in their original acquisition order, and quantitative results are calculated at the volume level.
>
> In our experiments, slices are extracted based on voxel resolution, with the lowest-resolution dimension chosen as the through-plane to preserve the highest image quality (as described in Section 2.3; we will clarify this further in the revised version). We note that this 2D approach may introduce through-plane inconsistencies, which we acknowledge as a limitation in the Discussion section. Extending DIST-CLIP to native 3D processing is therefore an important and a natural direction, and the framework itself is not limited to 2D: both the AST module and the disentanglement design naturally extend to 3D architectures. We are currently working on extending the model to 3D.
>
> ### Skull-Stripped Data
>
> In our experiments, we used skull-stripped images to reduce confounding variability and focus evaluation on contrast harmonization rather than non-brain structures. Importantly, DIST-CLIP does not assume skull stripping at the architectural level, and the disentanglement mechanism is not brain or preprocessing specific. We expect the method to generalize to non–skull-stripped data, although performance may be affected by additional anatomical variability. We acknowledge this as an important consideration and we have started training our framework with data with skull, and we will incorporate the results to the extension we are currently preparing.
>
> Regarding skull-stripped data and HACA3, we follow the original HACA3 implementation use their original weights and run it on images that include the skull. For quantitative metrics (PSNR and SSIM), we compute values only within the brain region for fair comparison and mask the brain for visualisation. We will make this clarification explicit in the revised manuscript.
>
> ### Lack of Downstream Task Evaluation
>
> We agree with the reviewer that evaluating DIST-CLIP on downstream tasks such as segmentation is essential for demonstrating its full clinical value. The primary objective of this work was to showcase our novel style transfer framework and establish the foundational correctness, flexibility, and anatomical fidelity of the contrast translation itself. By focusing on the disentanglement of anatomy and style, we ensure that our framework produces standardized, high-fidelity inputs that are optimized for practical diagnostic use. We are currently preparing an extension and will include these downstream evaluations, which will further illustrate the practical impact and robustness of our approach.
>
> ### Multi-Contrast Extension
>
> We agree that incorporating multi-contrast input is an important and promising direction. The current DIST-CLIP framework can be extended to leverage additional inputs, either by stacking anatomical images or through an attention-based mechanism. We plan to explore this idea in future work to further enhance the flexibility and utility of the model.
>
> ### T1 → T1 Evaluation
>
> T1 → T1 translation was not evaluated in the current experiments because our test set did not include patients with multiple T1-weighted acquisitions. We are actively extending the test set to enable more comprehensive evaluation across additional contrasts, which will be included in an extended version of the paper.

---

> ### Author Response · Authors · 2026-01-27
> **Thanks for increasing the rating**
>
> We thank the reviewer for the positive assessment and for reconsidering the rating in light of our rebuttal. We are glad that our clarifications addressed the concerns adequately and appreciate the reviewer’s view that the paper will be of interest to the conference audience.

---

### Comment · Area_Chair_7uWf · 2026-01-28
**Update final scores**

Hi reviewers

Can you please check if the authors' rebuttal addresses your concerns and update your final scores?

Thanks so much for your help in reviewing the paper!

Sincerely
AC

---

### Meta-Review · Area_Chair_7uWf · 2026-02-11

**Recommendation:** Accept (Oral)
**Confidence:** 5

**Metareview:**

The paper presents a novel approach to perform MRI harmonization by learning a joint representation of images and relevant clinical information contained in DICOM metadata using a disentangled style transfer based CLIP approach. As mentioned by reviewers', development of a 2D based approach is a limitation due to potential issues with inter-slice inconsistencies in the harmonization. It's understandable that the novelty is the technical development but the limitations regarding lack of evaluation in more realistic clinical scenarios with incomplete metadata and wider variations in imaging acquisitions should be acknowledged in discussion. Otherwise, this is an excellent paper.

---

### Decision · Program_Chairs · 2026-02-13

Accept (Poster)